# Identification of superior parental lines for biparental crossing via genomic prediction

**Ping-Yuan Chung, Chen-Tuo Liao** *

Department of Agronomy, National Taiwan University, Taipei, Taiwan

* ctliao@ntu.edu.tw

## Abstract

A parental selection approach based on genomic prediction has been developed to help plant breeders identify a set of superior parental lines from a candidate population before conducting field trials. A classical parental selection approach based on genomic prediction usually involves truncation selection, i.e., selecting the top fraction of accessions on the basis of their genomic estimated breeding values (GEBVs). However, truncation selection inevitably results in the loss of genomic diversity during the breeding process. To preserve genomic diversity, the selection of closely related accessions should be avoided during parental selection. We thus propose a new index to quantify the genomic diversity for a set of candidate accessions, and analyze two real rice (*Oryza sativa* L.) genome datasets to compare several selection strategies. Our results showed that the pure truncation selection strategy produced the best starting breeding value but the least genomic diversity in the base population, leading to less genetic gain. On the other hand, strategies that considered only genomic diversity resulted in greater genomic diversity but less favorable starting breeding values, leading to more genetic gain but unsatisfactorily performing recombination inbred lines (RILs) in progeny populations. Among all strategies investigated in this study, compromised strategies, which considered both GEBVs and genomic diversity, produced the best or second-best performing RILs mainly because these strategies balance the starting breeding value with the maintenance of genomic diversity.

## Introduction

Biparental crossing is a common scheme used for pure-line breeding in self-pollinated crops such as rice (*Oryza sativa* L.), wheat (*Triticum aestivum* L.), soybean (*Glycine max* [L.] Merr.), and oat (*Avena sativa* L.). Plant breeders cross two inbred parental lines to produce the $F_1$ population. Then, a subset of diverse individuals from the $F_2$ population is selected to produce potential recombination inbred lines (RILs) after several generations of selfing. Parental lines play a fundamental role in line development, and significantly affect the performance of the resulting RILs. However, the identification of superior parental lines from germplasm collections for maximizing selection response in subsequent cycles remains challenging for plant breeders [1, 2]. Another practical concern is that the number of possible crosses in such a breeding program is often far greater than what can be handled in the field. Therefore,

**Data Availability Statement:** All genotypic and phenotypic data used in this article are freely accessible and can be downloaded from the cited references.

**Funding:** This research was supported by the Ministry of Science and Technology, Taiwan (grant

number: MOST 108-2118-M-002-002-MY2). The funder had no role in study design, data collection and analysis, decision to publish, or preparation of the manuscript.

**Competing interests:** The authors declare that there is no conflict of interest.

**Abbreviations:** BLUP, best linear unbiased predictor; BRSW, brown rice seed width; FPP, florets per panicle; FT, flowering time; FTAA, flowering time at Arkansas; FTAF, flowering time at Faridpur; GBLUP, genomic best linear unbiased predicto; GD-O, algorithms considering genomic diversity only; GEBV, genomic estimated breeding value; GEBV-GD, algorithms considering both GEBV and genomic diversity; GEBV-O, algorithms considering GEBV only; PH, plant height; PNPP, panicle number per plant; RIL, recombinant inbred line; SNP, single nucleotide polymorphism; YLD, grain yield.

developing a method that can identify a limited number of superior parents before field trials would be of great help to plant breeders.

Genomic selection, based on the statistical method of genomic prediction (GP), has been used to improve breeding efficiency in dairy cattle [3] and a variety of crops [4–8]. The main concept of GP is to capture all the effects of quantitative trait loci (QTLs) using high-density DNA markers over the whole genome [9]. The most commonly used DNA markers are single nucleotide polymorphisms (SNPs). A GP model is first built using the phenotypic and genotypic data of a training population. Then, genomic estimated breeding values (GEBVs) for the candidate individuals with known genotypic data are predicted through the resulting GP model. Two kinds of mixed linear model methods are widely employed to obtain GEBVs: (i) best linear unbiased prediction (BLUP) based on markers, and (ii) BLUP based on a genomic relationship matrix. To perform marker-based BLUP, the marker effects are treated as random effects, and GEBVs of individuals are calculated by multiplying their marker scores by these BLUP estimates; the ridge regression BLUP (rr-BLUP) method [9, 10] follows this approach. To perform genomic relationship matrix-based BLUP, the genotypic values of individuals are treated as random effects and estimated through a genomic relationship matrix; this approach is used in the genomic BLUP (GBLUP) model [11, 12].

Gaynor et al. [13] proposed a two-part strategy for implementing genomic selection for line development, which addresses two components: (i) a product development component, to identify inbred lines either for hybrid parent development or cultivar release, and (ii) a population improvement component, to increase the frequency of favorable alleles through rapid recurrent genomic selection. Gaynor et al. [13] conducted a stochastic simulation and showed that programs using the two-part strategy generated up to 2.5- and 1.5-fold more genetic gain than conventional programs and the best performing standard genomic selection strategy, respectively. Additionally, Yao et al. [14] combined GP with Monte Carlo simulation to select superior parents in wheat breeding programs before field trials. The authors used the criterion of usefulness function on a selection index, which incorporates yield and two quality traits, to evaluate a cross, and concluded that the use of the usefulness function for parental selection resulted in higher genetic gain than the use of mid-parent GEBV, implying that the strategy for parental selection cannot only consider GEBVs of the candidate accessions.

By selecting parental lines with the highest GEBVs, breeders hope to maximally pass the favorable traits of parental lines on to their progeny. However, the truncation selection approach risks the elimination of several favorable QTLs from the breeding population because of a lack of genomic diversity [15]. Therefore, in this study, we took both GEBV and genomic diversity into account for identifying superior parents in a biparental crossing program. We constructed a GBLUP model for a specific target trait to predict the GEBVs of the candidate accessions. We first proposed a new index to quantify the genomic diversity of a set of candidate accessions. Subsequently, we simulated the genotypic data for progeny populations derived from a cross over successive generations, and predicted the GEBVs of the simulated progeny populations through the trained GBLUP model. Then, we made generation advancement decisions according to the resulting GEBVs. Finally, we assessed a set of parental lines based on $F_{10}$ RILs. We compared the performance of several selection strategies via analysis of two real rice genome datasets.

## Materials and methods

### Rice genome datasets

**Dataset I.** The rice genome dataset originally collected for genome-wide association study (GWAS) in Zhao et al. [16] was used to illustrate the proposed procedure. This dataset

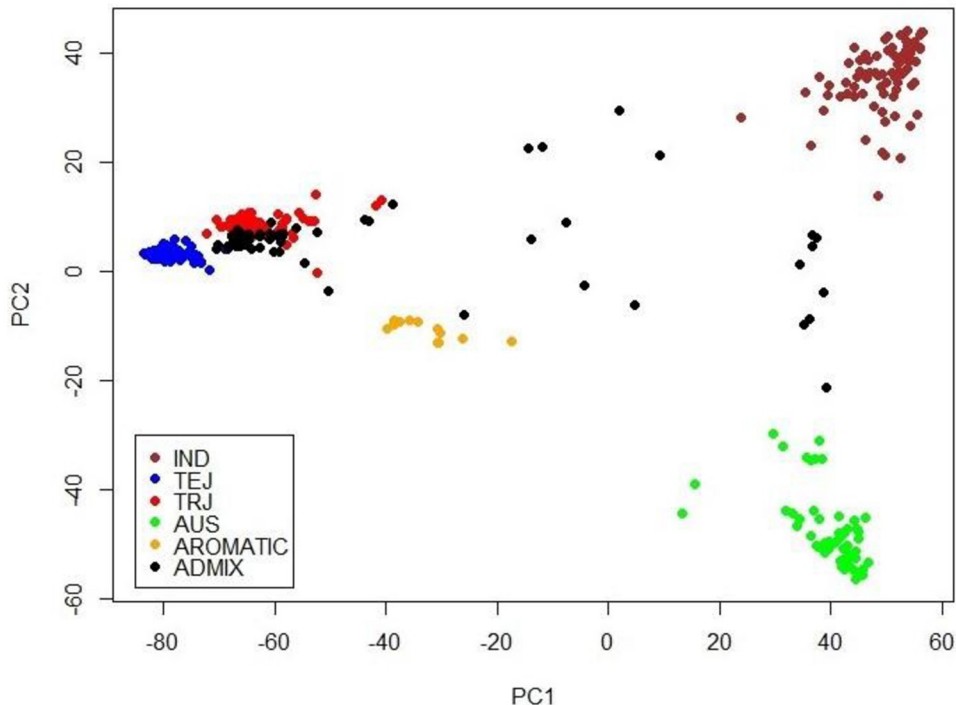

**Fig 1. Scatter plot of 413 accessions with 11,047 SNPs according to the first two principal components (PCs) for the 44k rice genome dataset.** IND: *indica* rice; TEJ: temperate *japonica* rice; TRJ: tropical *japonica* rice; AUS: Aus rice; AROMATIC: aromatic rice; ADMIX: admixed group.

contains 44,100 SNP variants and 36 traits of 413 *O. sativa* accessions, which comprises five subpopulations and one admixed group. SNPs with missing rate > 0.05 and minor allele frequency < 0.05 were removed from the dataset. To reduce redundant collinearity in the genomic relationship matrix, one SNP was randomly selected from each bin of 20,000 bp over each chromosome. A scatter plot based on the first two principal components (PCs) using the retained 11,047 SNPs is displayed in Fig 1, which is almost the same as that in the corresponding plot generated using all 44,100 SNPs [16]. The SNP genotype at each locus was coded as -1, 0, or 1, where 1 indicates homozygous genotype of the major allele; -1 indicates homozygous genotype of the minor allele; and 0 indicates heterogenous genotype. After SNP coding, any missing locus was imputed as 1. Six traits were analyzed: brown rice seed width (BRSW), florets per panicle (FPP), flowering time at Arkansas (FTAA), flowering time at Faridpur (FTAF), plant height (PH), and panicle number per plant (PNPP).

**Dataset II.** The rice genome dataset, which was collected for genomic selection study [8], was further analyzed as dataset II. This dataset contains 73,147 SNP variants and 363 elite breeding lines belonging to *indica* or *indica*–admixed group. Phenotypic observations include four years (2009–2012; two seasons per year [dry and wet]) of data on grain yield (YLD), flowering time (FT), and plant height (PH), although PH data in the wet season of 2009 were not available. Phenotypic values of 35 out of 363 individuals were missing; therefore, adjusted means of only 328 individuals were used in this study. Additionally, only 10,772 out of 73,147 SNPs were used in this study. One SNP marker was selected per 0.1-cM interval on each chromosome because the chosen subset of the full marker set has been shown to be efficient enough for genomic selection in this collection of rice germplasm [8].

## Monte Carlo simulation of the genotypic data of progeny populations

To simulate the genotypic data of progeny populations, the Gramene Annotated Nipponbare Sequence [17] was used to estimate recombination rates between two adjacent SNPs. The Gramene Annotated Nipponbare Sequence database contains both physical and linkage distances between SNPs, which can be downloaded from http://archive.gramene.org. The genetic positions of SNPs were estimated via linear interpolation between the two markers flanking each SNP. Once the genetic positions were obtained, the recombination rates between adjacent SNPs were estimated using Haldane's mapping function [18]:

$$r_{AB} = \frac{1}{2}(1 - e^{-2X_{AB}})$$

where $r_{AB}$ is the recombination rate between markers A and B; $X_{AB}$ is the linkage distance between markers A and B; and $e$ is Euler's number, a mathematical constant approximately equal to 2.71828. Based on a series of Bernoulli distributions and the estimated recombination rates, the crossover of each chromosome was simulated to yield the sequence of a gamete. Then, two gametes were paired to produce the genotypic data for the progeny.

## GBLUP model

The following GBLUP model was considered for GP:

$$y = \mu \mathbf{1}_n + g + e \tag{1}$$

where **y** denotes the vector of phenotypic values of a training population with $n$ individuals; $\mu$ is a constant term; $\mathbf{1}_n$ is the vector of order $n$ with all elements equal to 1; $g$ represents the vector of genotypic values; and $e$ is the vector of random errors.

It is assumed that $g$ follows a multivariate normal distribution $MVN(\mathbf{0}, \sigma_g^2 K)$, where $\mathbf{0}$ is a zero vector; $\sigma_g^2$ is the genetic variance of additive effects; and $K$ is a genomic relationship matrix among the individuals. Furthermore, $e$ follows $MVN(\mathbf{0}, \sigma_e^2 I_n)$, where $\sigma_e^2$ is the random error variance, and $I_n$ denotes the identity matrix of order $n$. Here, $g$ and $e$ are assumed to be mutually independent. In this study, the genomic relationship matrix $K = MM^T/p$ was considered, where $M$ is the marker score matrix, and $p$ is the number of SNP markers.

The model parameters of the GBLUP model can be estimated through the Henderson's equation [19], as follows:

$$\begin{bmatrix} n & \mathbf{1}_n^T \\ \mathbf{1}_n & I_n + \lambda K^{-1} \end{bmatrix} \begin{bmatrix} \hat{\mu} \\ \hat{g} \end{bmatrix} = \begin{bmatrix} \mathbf{1}_n^T y \\ y \end{bmatrix} \tag{2}$$

where the regularization parameter $\lambda$ is given by $\lambda = \frac{\sigma_e^2}{\sigma_g^2}$.

The mmer () function in the R package sommer [20] was used to obtain the restricted maximum likelihood estimates (REMLs) for the two variance components of $\sigma_g^2$ and $\sigma_e^2$, and the resulting estimates were entered into Eq (2) to obtain $\hat{\mu}$ and $\hat{g}$.

If $\hat{g}_{bp}$ is considered as the vector of estimated genotypic values for a breeding population, and $K_{bp}$ is considered as the genomic relationship matrix between the breeding and training populations, the following equation is obtained:

$$\hat{g}_{bp} = K_{bp}K^{-1}\hat{g}$$

The GEBV for the breeding population is $\hat{g}_{bp}$ plus the estimate of the constant term $\hat{\mu}$.

## Index for quantifying genomic diversity

Let $g_0$ be the vector of genotypic values, and $K_0$ be the genomic relationship matrix for a particular set of accessions with size $n_0$. According to the GBLUP model in Eq (1), the covariance matrix for $g_0$ is given by:

$$\mathrm{Var}(g_0) = \sigma_g^2 K_0$$

The determinant of the covariance matrix represents the overall variability of the genotypic values, which is calculated as:

$$|\mathrm{Var}(g_0)| = (\sigma_g^2)^{n_0}|K_0| \qquad (3)$$

Clearly, the determinant of Eq (3) is proportional to the $D$-score defined below:

$$D\text{-score} = |K_0| \qquad (4)$$

For a fixed number of $n_0$, a subset of accessions chosen from a breeding population with the maximal $D$-score will have greater genomic diversity than the competing choices with size $n_0$. The concept of the $D$-score is adopted from optimum experimental designs [21].

A simple example is given below to illustrate the $D$-score. Suppose that there are three accessions ($n = 3$) in the candidate set with the genomic relationship matrix:

$$K = \begin{bmatrix} 1 & 0.9 & 0.5 \\ 0.9 & 1 & 0.1 \\ 0.5 & 0.1 & 1 \end{bmatrix}$$

For $n_0 = 2$, the $D$-score for $g_1$ and $g_2$ is calculated as $|K_0| = \begin{vmatrix} 1 & 0.9 \\ 0.9 & 1 \end{vmatrix} = 0.19$. Similarly, the $D$-scores for $g_1$ and $g_3$ and for $g_2$ and $g_3$ are given as 0.75 and 0.99, respectively. Clearly, the two accessions with $g_2$ and $g_3$ genotypic values have greater genomic variation (smaller genomic correlation) than the other competing choices. Closely related individuals could be excluded from the maximal $D$-score set. The genetic algorithm presented in Ou and Liao [22] was used to search a subset of accessions from a candidate population, such that it can attain the maximal $D$-score of Eq (4).

## Procedure for selecting parental lines

To evaluate a variety of strategies for determining parental lines, the following steps were carried out:

1. For a specific target trait, all phenotypic values available from each rice genome dataset were used to build the corresponding GBLUP model of Eq (1).

2. The GEBVs of all accessions in the dataset were predicted through the trained GBLUP model in step 1. Seven strategies were used to select a subset of 10 parental lines according to their GEBVs: (i) the GEBV only (GEBV-O) approach, which chose the top 10 accessions (either maximal or minimal); three genomic diversity only (GD-O) approaches: (ii) GD-O-30, (iii) GD-O-50, and (iv) GD-O-100, which applied the genetic exchange algorithm to search for an optimal subset of 10 accessions from each of the three candidate sets composed of the top 30, 50, and 100 accessions, respectively, such that the chosen subset had the maximal $D$-score; and three approaches that considered both GEBV and genomic diversity: (v) GEBV-GD-30, (vi) GEBV-GD-50, and (vii) GEBV-GD-100, which retained the top

two accessions, and then applied the genetic exchange algorithm to search for another eight accessions from the remainder of each candidate set for GD-O-30, GD-O-50, and GD-O-100, respectively, so that the resulting 10 accessions had the maximal $D$-score.

3. For each subset of 10 accessions determined by the seven strategies, any two parental lines were crossed to produce 45 $F_1$ hybrids. Here, we started to simulate the genotypic data for successive generations of progeny populations through the Monte Carlo simulation. Each of the 45 $F_1$ hybrids produced 60 individuals of the $F_2$ population by self-pollination, resulting in 2,700 $F_2$ individuals. After obtaining the GEBVs for the 2,700 $F_2$ individuals via the trained GBLUP model, the top 45 $F_2$ individuals were retained. Again, these 45 $F_2$ individuals were used to produce 2,700 $F_3$ individuals (60 $F_3$ individuals per $F_2$ individual), and the top 45 $F_3$ individuals were retained. The same procedure was repeated to produce 2,700 $F_{10}$ individuals, which were assumed to be a fixed population.

4. For the resulting 2,700 $F_{10}$ individuals generated according to each strategy, we found the best $F_{10}$ RILs with the highest GEBVs.

A flowchart of this procedure is displayed in Fig 2. This procedure was repeated 30 times to obtain the best $F_{10}$ RILs from each repetition for each strategy. The average of the GEBVs for the best $F_{10}$ RILs was then calculated and used as the measure of efficiency for the strategy. Then, pairwise comparisons were performed among the GEBV averages, based on the least significant difference (LSD) test. Note that for BRSW, FPP, and PNPP in Dataset I and for YLD in Dataset II, larger GEBVs are preferable (i.e., for these traits, the larger the GEBV, the better). The remaining five traits (FTAA, FTAF, and PH in Dataset I, and FT and PH in Dataset II) follow the rule: that the smaller the GEBV, the better.

## Calculation of genetic gain

To understand the genetic improvement in a target trait using different strategies, the genetic gain was estimated as:

$$\text{Genetic gain} = \overline{GEBV}_{F_{10}} - \overline{GEBV}_P \tag{5}$$

where $\overline{GEBV}_{F_{10}}$ denotes the average GEBV of the resulting 2,700 $F_{10}$ RILs; and $\overline{GEBV}_P$ denotes the average GEBV of 10 parental lines selected using each strategy [23]. The larger the absolute value of the genetic gain, the greater the improvement in the target trait. The average of genetic gains from 30 repetitions was reported for each strategy, and multiple comparisons among the genetic gain averages were performed using the LSD test.

# Results

## Comparison of strategies based on the best $F_{10}$ RILs

The GEBV averages of the best $F_{10}$ RILs and results of the LSD test are displayed in Tables 1 and 2 for Datasets I and II, respectively. The strategies that considered both GEBV and genomic diversity (GEBV-GD-30, -50, -100) generally showed satisfactory efficiency because they achieved the best or second-best performance for all traits. Therefore, these types of strategies could be used as a reliable means for selecting parental lines. On the other hand, strategies accounting for only genomic diversity (GD-O-30, -50, -100) did not show satisfactory efficiency for all traits, except GD-O-100, which was satisfactory for YLD in Dataset II. The GEBV-O strategy showed the best or second-best performance for FPP and PH in Dataset I and for PH and FT in Dataset II, but it also showed the worst or second-worst performance for the remaining four traits in Dataset I and for YLD in Dataset II. These data indicate that

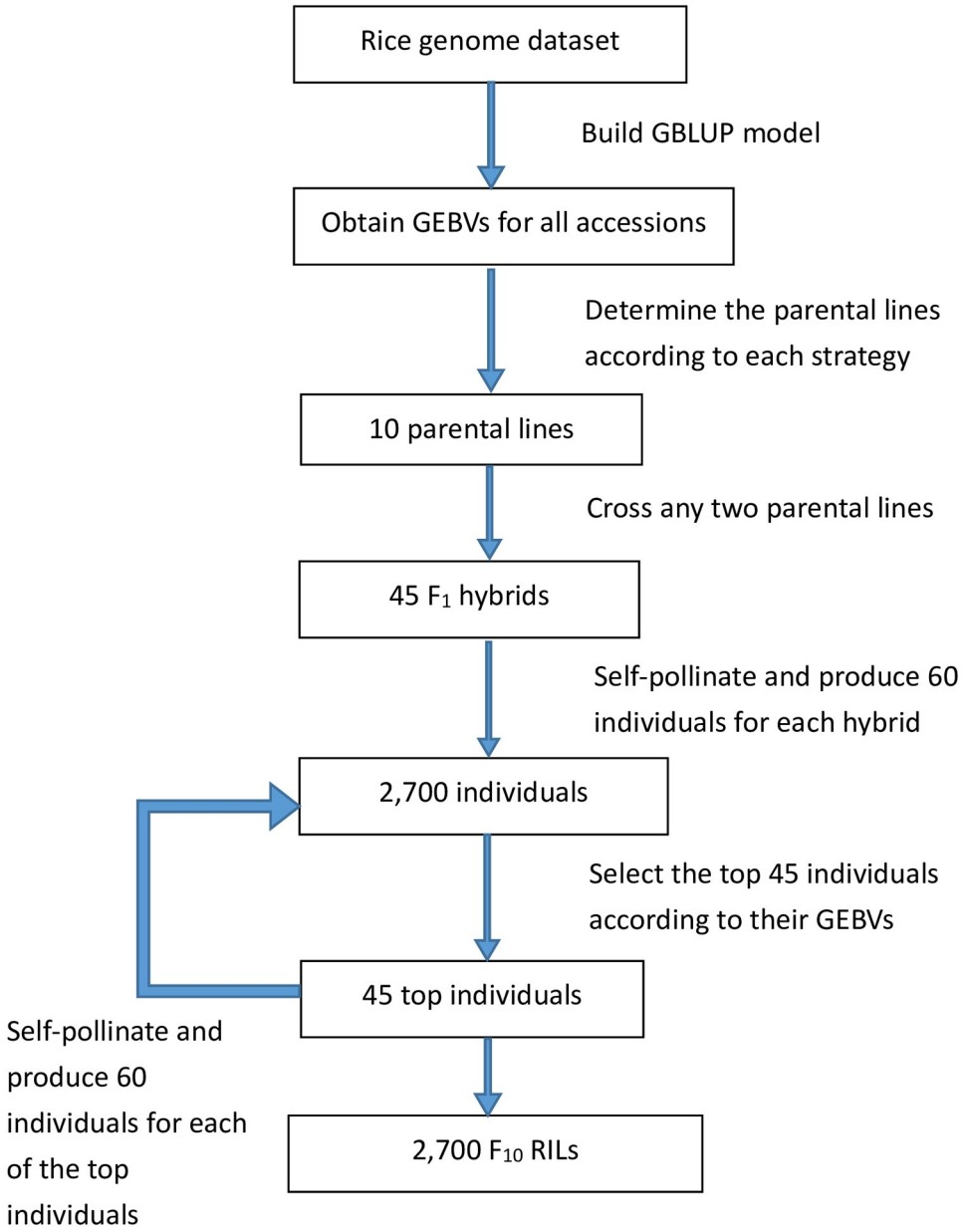

**Fig 2. Flowchart showing the Monte Carlo simulation.** GEBV: genomic estimated breeding value; GBLUP: genomic best linear unbiased predictor; RIL: recombinant inbred line.

GEBV-O is a high-risk strategy. In general, the results of the LSD test showed significant differences in GEBV averages between the best/second-best and worst/second-worst performances for all traits in both datasets.

We also displayed the average GEBV ± standard deviation (SD) of the best RILs selected by 30 repetitions over consecutive generations in Figs 3, 4 and 5. Four strategies including GEBV-O, GEBV-GD-30, -50, and -100 selected the same best individual from the 30 repetitions at the parental generation and also at the $F_1$ generation; therefore, no SD is shown with the corresponding GEBV averages. The GEBV averages of the best parental lines selected by the strategies can be ranked as GEBV-O = GEBV-GD-30 = GEBV-GD-50 = GEBV-GD-

**Table 1. Ranking and GEBV averages (in parentheses) of the best $F_{10}$ RILs selected by 30 repetitions of the seven proposed strategies applied to six traits in Dataset I.**

| Strategy[1] | Traits[2] | | | | | |
|---|---|---|---|---|---|---|
| | **BRSW** | **FPP** | **FTAA** | **FTAF** | **PH** | **PNPP** |
| GEBV-O | 6 (3.418)e | **2 (5.961)a** | 6 (56.52)e | 6 (61.85)d | **1 (42.18)a** | 6 (4.125)c |
| GD-O-30 | 7 (3.408)e | 5 (5.951)a | 3 (51.56)b | 3 (59.35)a | 5 (49.33)b | 3 (4.188)b |
| GD-O-50 | 3 (3.576)c | 6 (5.916)b | 5 (53.34)d | 5 (60.12)c | 6 (49.80)b | 5 (4.138)c |
| GD-O-100 | 4 (3.496)d | 7 (5.882)c | 7 (56.83)e | 7 (61.96)d | 7 (51.78)c | 7 (4.086)d |
| GEBV-GD-30 | 5 (3.419)e | 3 (5.954)a | **1 (47.13)a** | **1 (59.21)a** | **2 (42.69)a** | **1 (4.225)a** |
| GEBV-GD-50 | **1 (3.656)a** | **1 (5.964)a** | **2 (47.45)a** | **2 (59.30)a** | 3 (43.23)a | **2 (4.214)a** |
| GEBV-GD-100 | **2 (3.634)b** | 4 (5.953)a | 4 (51.38)c | 4 (59.63)b | 4 (43.49)a | 4 (4.171)b |

[1] GEBV-O: subset of the top 10 accessions with minimal or maximal GEBV; GD-O-30, -50, -100: subsets of 10 accessions with the maximal D-scores chosen from candidate sets comprising the top 30, 50, and 100 accessions, respectively; GEBV-GD-30, -50, -100: subsets of the top two accessions plus eight accessions chosen from the remainder of the candidate sets composed of the top 30, 50, and 100 accessions, respectively, with the maximal D-scores.

[2] BRSW: brown rice seed width; FPP: florets per panicle; FTAA: flowering time at Arkansas; FTAF: flowering time at Faridpur; PH: plant height; PNPP: panicle number per plant. Different lowercase letters indicate significant differences among the strategies for a given trait ($P < 0.01$; LSD test). The best and second-best strategies are indicated in bold, while the worst and second-worst strategies are underlined.

$100 \geq$ GD-O-30 $\geq$ GD-O-50 $\geq$ GD-O-100 in decreasing desirability. The desirability at the parental generation decreased as the degree of diversity increased for the three strategies, considering the genomic diversity only. Additionally, the desirability declined from the parental generation to $F_1$ generation for every strategy because of the heterozygous alleles in $F_1$ hybrids.

To explore the extent to which the top two accessions contributed to the subset of 10 parental lines determined by four strategies (GEBV-O, GEBV-GD-30, -50, and -100), we compared each subset with a reduced group consisting of $F_1$ hybrids, whose parental lines contained at least one of the top two accessions for each subset. Each reduced group consisted of 17 $F_1$

**Table 2. Ranking and GEBV averages (in parentheses) of the best $F_{10}$ RILs selected by 30 repetitions of the seven proposed strategies applied to three traits in Dataset II.**

| Strategy[1] | Traits[2] | | |
|---|---|---|---|
| | **YLD** | **PH** | **FT** |
| GEBV-O | 7 (6472)b | **1 (85.817)a** | **2 (77.818)a** |
| GD-O-30 | 4 (6491)b | 5 (87.517)b | 7 (78.410)c |
| GD-O-50 | 5 (6489)b | 6 (89.920)c | 5 (78.164)b |
| GD-O-100 | **1 (6546)a** | 7 (91.799)e | 6 (78.359)bc |
| GEBV-GD-30 | 3 (6506)ab | 3 (85.976)a | 4 (77.883)a |
| GEBV-GD-50 | 6 (6485)b | **2 (85.917)a** | **1 (77.725)a** |
| GEBV-GD-100 | **2 (6539)a** | 4 (86.062)a | 3 (77.873)a |

[1] GEBV-O: subset of the top 10 accessions with the minimal or maximal GEBV; GD-O-30, -50, -100: subsets of 10 accessions with maximal D-scores chosen from the candidate sets comprising the top 30, 50, and 100 accessions, respectively; GEBV-GD-30, -50, -100: subsets of the top two accessions plus eight accessions chosen from the remainder of the candidate sets composed of the top 30, 50, and 100 accessions, respectively, with the maximal D-scores.

[2] YLD: yield; PH: plant height; FT: flowering time. Different lowercase letters indicate significant differences among the strategies for a given trait ($P < 0.01$; LSD test). The best and second-best strategies are indicated in bold, while the worst and second-worst strategies are underlined.

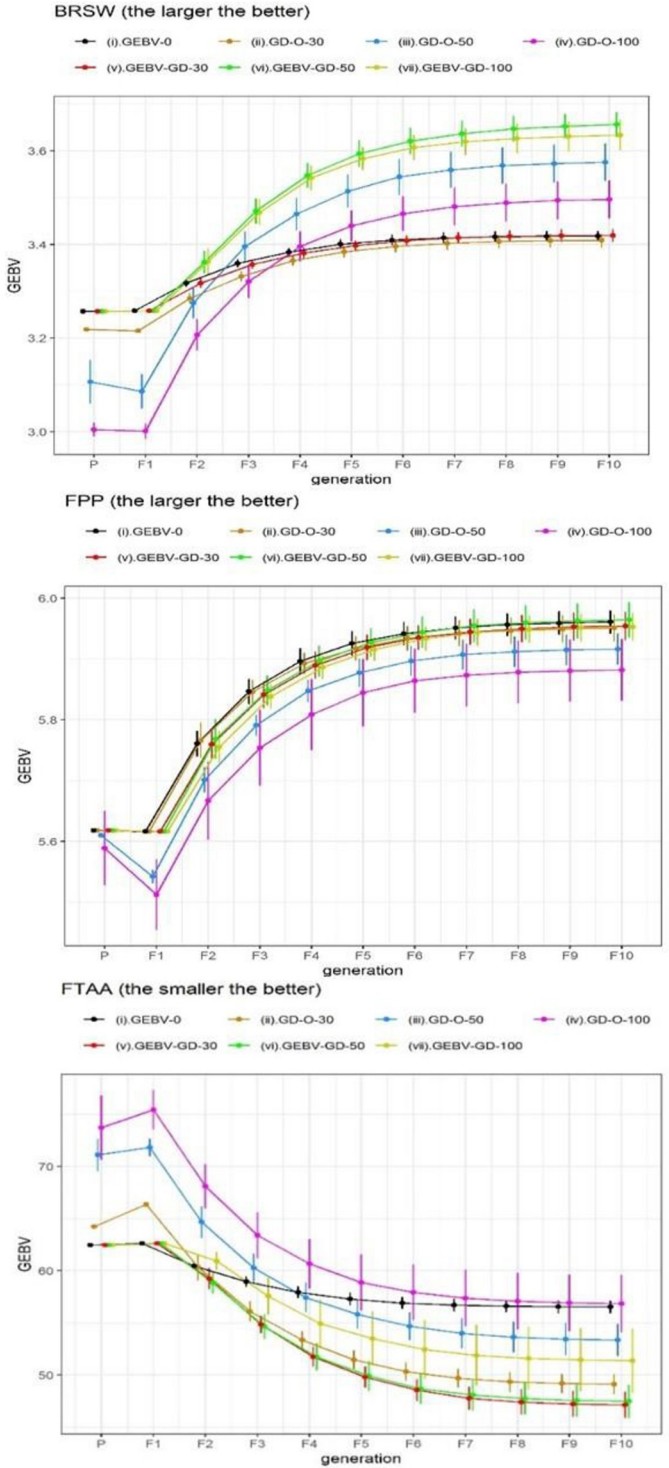

**Fig 3. GEBV averages of the best individuals selected from 30 repetitions at consecutive generations for the six chosen traits in Dataset I.** GEBV-O: subset of the top 10 accessions with minimal or maximal GEBVs; GD-O-30, -50, -100: subsets of 10 accessions with maximal D-scores chosen from candidate sets composed of the top 30, 50, and 100 accessions, respectively; GEBV-GD-30, -50, -100: subsets of the top two accessions plus eight accessions chosen from the remainder of the candidate sets composed of the top 30, 50, and 100 accessions, respectively, with maximal D-scores; BRSW: brown rice seed width; FPP: florets per panicle; FTAA: flowering time at Arkansas.

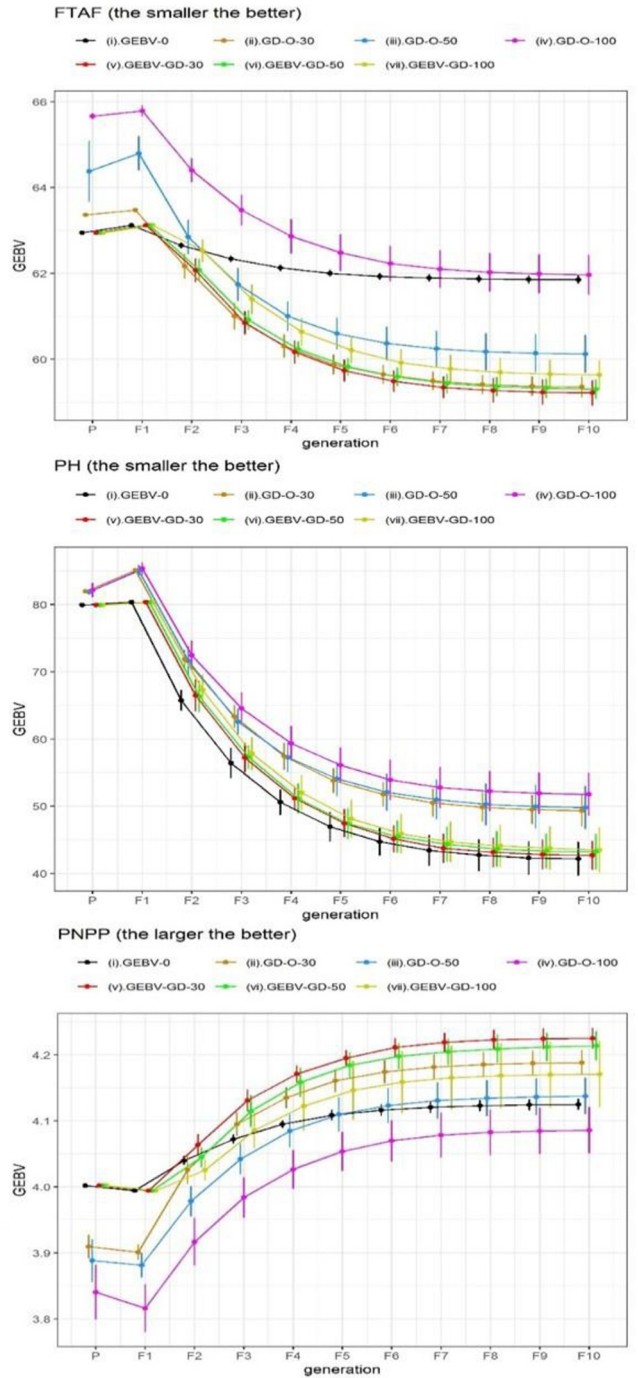

**Fig 4. GEBV averages of the best individuals selected from 30 repetitions at consecutive generations for the six chosen traits in Dataset I.** GEBV-O: subset of the top 10 accessions with minimal or maximal GEBVs; GD-O-30, -50, -100: subsets of 10 accessions with maximal D-scores chosen from candidate sets composed of the top 30, 50, and 100 accessions, respectively; GEBV-GD-30, -50, -100: subsets of the top two accessions plus eight accessions chosen from the remainder of the candidate sets composed of the top 30, 50, and 100 accessions, respectively, with maximal D-scores; FTAF: flowering time at Faridpur; PH: plant height; PNPP: panicle number per plant.

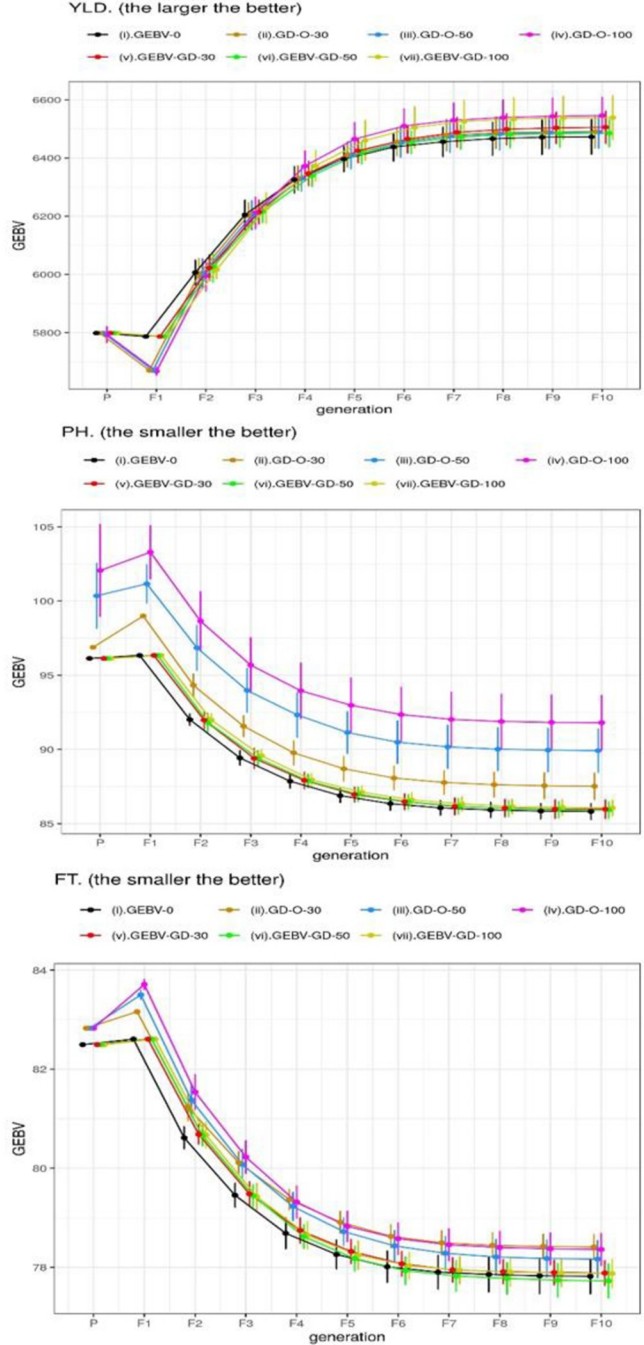

**Fig 5. GEBV averages of the best individuals selected from 30 repetitions at consecutive generations for the three target traits in Dataset II.** GEBV-O: subset of the top 10 accessions with minimal or maximal GEBVs; GD-O-30, -50, -100: subsets of 10 accessions with maximal D-scores chosen from candidate sets composed of the top 30, 50, and 100 accessions, respectively; GEBV-GD-30, -50, -100: subsets of the top two accessions plus eight accessions chosen from the remainder of the candidate sets composed of the top 30, 50, and 100 accessions, respectively, with the maximal D-scores; YLD, yield; PH, plant height; FT, flowering time.

**Table 3. GEBV averages of the best $F_{10}$ RILs selected by 30 repetitions, based on the original group comprising 45 $F_1$ hybrids and the reduced group comprising 17 $F_1$ hybrids, using four strategies.**

| Trait[1] | Strategy[2] | | | | | | | |
|---|---|---|---|---|---|---|---|---|
| | GEBV-O | | GEBV-GD-30 | | GEBV-GD-50 | | GEBV-GD-100 | |
| Dataset I | 45 $F_1$ | 17 $F_1$ | 45 $F_1$ | 17 $F_1$ | 45 $F_1$ | 17 $F_1$ | 45 $F_1$ | 17 $F_1$ |
| BRSW | 3.418 | 3.423 | 3.419 | 3.418 | 3.656 | 3.652 | 3.634 | 3.650 |
| FPP | 5.961 | 5.965 | 5.954 | 5.957 | 5.964 | 5.958 | 5.953 | 5.943 |
| FTAA | 56.521 | 57.513 | 47.136 | 46.961 | 47.457 | 47.421 | 51.382 | 51.734 |
| FTAF | 61.856 | 61.850 | 59.216 | 59.123 | 59.304 | 59.232 | 59.634 | 59.713 |
| PH | 42.185 | 43.409 | 42.699 | 43.271 | 43.232 | 43.791 | 43.498 | 43.854 |
| PNPP | 4.125 | 4.129 | 4.225 | 4.226 | 4.214 | 4.204 | 4.171 | 4.161 |
| Dataset II | 45 $F_1$ | 17 $F_1$ | 45 $F_1$ | 17 $F_1$ | 45 $F_1$ | 17 $F_1$ | 45 $F_1$ | 17 $F_1$ |
| YLD | 6472 | 6476 | 6506 | 6499 | 6485 | 6484 | 6539 | 6534 |
| PH | 85.817 | 85.991 | 85.976 | 85.844 | 85.917 | 86.092 | 86.062 | 86.060 |
| FT | 78.818 | 77.834 | 77.883 | 77.750 | 77.725 | 77.778 | 77.873 | 77.690 |

[1] BRSW: brown rice seed width; FPP: florets per panicle; FTAA: flowering time at Arkansas; FTAF: flowering time at Faridpur; PH: plant height; PNPP: panicle number per plant; YLD: yield; PH: plant height; FT: flowering time.

[2] GEBV-O: subset of the top 10 accessions with minimal or maximal GEBVs; GEBV-GD-30, -50, -100: subsets of the top two accessions plus eight accessions chosen from the remainder of the candidate sets composed of the top 30, 50, and 100 accessions, respectively, with the maximal D-scores.

hybrids. Similarly, we followed the analysis procedure to obtain the GEBV averages of the best $F_{10}$ RILs from 30 repetitions based on the reduced group. The results are displayed in Table 3, with the corresponding GEBV averages based on the group of the original 45 $F_1$ hybrids. The results showed no practical significance between these two groups for all the traits using the four strategies (Table 3). Therefore, the reduced group can be an alternative to the full group.

## Genetic gains with different strategies

The average genetic gains and results of the LSD test are displayed in Tables 4 and 5 for Datasets I and II, respectively. It is also reasonable to evaluate the performance of the strategies according to the endpoint of $\overline{GEBV}_{F_{10}}$. The comparison results based on $\overline{GEBV}_{F_{10}}$ were consistent with the above comparison results based on the best $F_{10}$ RILs. Strategies that considered genomic diversity (GD-O-30, -50, -100; GEBV-GD-30, -50, -100) showed greater genetic gain than the GEBV-O for all traits, except PH in Dataset I (Table 4). The genetic gain generally increased with the increase in genomic diversity, as expected (GD-O-100 outperformed both GD-O-50 and GD-O-30 for all traits, except BRSW and FTAF in Dataset I; GEBV-GD-100 outperformed both GEBV-GD-50 and GEBV-GD-30 for all traits). The results of the LSD test showed that the GEBV-GD-100 strategy significantly differs from the remaining strategies in genetic gain for all traits in Dataset I, but it showed no significant difference from GEBV-GD-50 for FTAA and from GEBV-GD-50 and -30 for PH. On the other hand, the GD-O-100 strategy significantly differed from the remaining strategies for all traits in Dataset II, except from the GEBV-GD-100 for PH. In addition, GEBV-O showed the best $\overline{GEBV}_P$, while GEBV-GD-30, -50, and -100 showed higher $\overline{GEBV}_P$ than their counterparts (GD-O-30, -50, and -100, respectively) for all traits in both datasets. Thus, a strategy has a relatively good starting point, as it considers more candidate accessions with the top GEBVs.

**Table 4. Average genetic gains from 30 repetitions for Dataset I.**

| Strategy[1] | Brown rice seed width (BRSW) | | | Florets per panicle (FPP) | | |
|---|---|---|---|---|---|---|
| | $\overline{GEBV}_P$ [2] | $\overline{GEBV}_{F_{10}}$ [3] | Genetic gain[4] | $\overline{GEBV}_P$ | $\overline{GEBV}_{F_{10}}$ | Genetic gain |
| GEBV-O | 3.17 | 3.42 | 0.25f | 5.51 | 5.96 | 0.45f |
| GD-O-30 | 3.10 | 3.41 | 0.31e | 5.48 | 5.95 | 0.47e |
| GD-O-50 | 3.00 | 3.57 | 0.57c | 5.41 | 5.91 | 0.50d |
| GD-O-100 | 2.94 | 3.49 | 0.55d | 5.31 | 5.88 | 0.57b |
| GEBV-GD-30 | 3.12 | 3.42 | 0.30e | 5.48 | 5.95 | 0.47e |
| GEBV-GD-50 | 3.04 | 3.65 | 0.61b | 5.43 | 5.96 | 0.53c |
| GEBV-GD-100 | 3.00 | 3.63 | 0.63a | 5.34 | 5.95 | 0.61a |
| | Flowering time at Arkansas (FTAA) | | | Flowering time at Faridpur (FTAF) | | |
| | $\overline{GEBV}_P$ | $\overline{GEBV}_{F_{10}}$ | Genetic gain | $\overline{GEBV}_P$ | $\overline{GEBV}_{F_{10}}$ | Genetic gain |
| GEBV-O | 64.30 | 56.57 | -7.73d | 63.45 | 61.87 | -1.58f |
| GD-O-30 | 72.25 | 49.26 | -22.99bc | 64.93 | 59.40 | -5.53cd |
| GD-O-50 | 75.41 | 53.54 | -21.87c | 65.82 | 60.16 | -5.66c |
| GD-O-100 | 80.01 | 57.00 | -23.01bc | 67.34 | 62.01 | -5.33e |
| GEBV-GD-30 | 71.09 | 47.31 | -23.78b | 64.68 | 59.25 | -5.43de |
| GEBV-GD-50 | 72.86 | 47.64 | -25.22a | 65.40 | 59.35 | -6.05b |
| GEBV-GD-100 | 77.16 | 51.53 | -25.63a | 66.46 | 59.68 | -6.78a |
| | Plant height (PH) | | | Panicle number per plant (PNPP) | | |
| | $\overline{GEBV}_P$ | $\overline{GEBV}_{F_{10}}$ | Genetic gain | $\overline{GEBV}_P$ | $\overline{GEBV}_{F_{10}}$ | Genetic gain |
| GEBV-O | 83.77 | 42.52 | -41.25b | 3.93 | 4.12 | 0.19e |
| GD-O-30 | 89.50 | 49.69 | -39.81b | 3.86 | 4.19 | 0.33d |
| GD-O-50 | 90.11 | 50.13 | -39.98b | 3.80 | 4.14 | 0.34d |
| GD-O-100 | 92.10 | 52.10 | -40.00b | 3.64 | 4.08 | 0.44b |
| GEBV-GD-30 | 87.26 | 42.99 | -44.27a | 3.90 | 4.22 | 0.32d |
| GEBV-GD-50 | 87.95 | 43.50 | -44.45a | 3.84 | 4.21 | 0.37c |
| GEBV-GD-100 | 89.27 | 43.95 | -45.32a | 3.70 | 4.17 | 0.47a |

[1] GEBV-O: subset of the top 10 accessions with minimal or maximal GEBVs; GD-O-30, -50, -100: subsets of 10 accessions with maximal D-scores chosen from the candidate sets composed of the top 30, 50, and 100 accessions, respectively; GEBV-GD-30, -50, -100: subsets of the top two accessions plus eight accessions chosen from the remainder of the candidate sets composed of the top 30, 50, and 100 accessions, respectively, with the maximal D-scores.

[2] $\overline{GEBV}_P$: average GEBV of the 10 selected parental lines.

[3] $\overline{GEBV}_{F_{10}}$: average GEBV of 2,700 $F_{10}$ RILs.

[4] Lowercase letters indicate significant differences among strategies for a given trait ($P < 0.01$; LSD test).

## Discussion

Dataset II was specifically collected for genomic selection. All the available accessions in this dataset are *indica* or *indica*–admixed. The results of performance based on the best $F_{10}$ RILs (Table 2) revealed that all seven strategies showed similar performance for the three target traits. The resulting GEBV averages of the best $F_{10}$ RILs ranged from 6472 to 6546 kg/ha for YLD, from 85.889 to 91.852 cm for PH, and from 77.725 to 78.410 days for FT. This could be because the candidate accessions in Dataset II are elite breeding lines, with limited genetic diversity and similar phenotypic values for the target traits. However, the results of the LSD test revealed that the two strategies (GD-O-100 and GEBV-GD-100) with greater genomic diversity for YLD led to significantly larger YLD than the other five strategies. Four strategies

**Table 5. Average genetic gains derived from 30 repetitions for Dataset II.**

| Strategy[1] | Yield (YLD) | | |
|---|---|---|---|
| | $\overline{GEBV}_P$ [2] | $\overline{GEBV}_{F_{10}}$ [3] | Genetic gain[4] |
| GEBV-O | 5571.61 | 6468.60 | 896.99e |
| GD-O-30 | 5452.39 | 6488.02 | 1035.63c |
| GD-O-50 | 5436.58 | 6484.58 | 1048.00bc |
| GD-O-100 | 5289.74 | 6540.72 | 1250.98a |
| GEBV-GD-30 | 5538.44 | 6501.23 | 962.79d |
| GEBV-GD-50 | 5522.45 | 6482.13 | 959.68d |
| GEBV-GD-100 | 5454.37 | 6535.79 | 1081.42b |
| | Plant height (PH) | | |
| | $\overline{GEBV}_P$ | $\overline{GEBV}_{F_{10}}$ | Genetic gain |
| GEBV-O | 97.75 | 85.89 | -11.86d |
| GD-O-30 | 102.20 | 87.59 | -14.61a |
| GD-O-50 | 103.66 | 89.99 | -13.67b |
| GD-O-100 | 106.83 | 91.85 | -14.98a |
| GEBV-GD-30 | 99.00 | 86.01 | -12.99c |
| GEBV-GD-50 | 99.39 | 85.99 | -13.40bc |
| GEBV-GD-100 | 101.15 | 86.13 | -15.02a |
| | Flowering time (FT) | | |
| | $\overline{GEBV}_P$ | $\overline{GEBV}_{F_{10}}$ | Genetic gain |
| GEBV-O | 83.14 | 77.84 | -5.30e |
| GD-O-30 | 83.98 | 78.43 | -5.55d |
| GD-O-50 | 84.57 | 78.19 | -6.38b |
| GD-O-100 | 85.62 | 78.39 | -7.23a |
| GEBV-GD-30 | 83.44 | 77.90 | -5.54d |
| GEBV-GD-50 | 83.69 | 77.76 | -5.93c |
| GEBV-GD-100 | 84.16 | 77.89 | -6.27b |

[1] GEBV-O: subset of the top 10 accessions with minimal or maximal GEBVs; GD-O-30, -50, -100: subsets of 10 accessions with maximal D-scores chosen from the candidate sets composed of the top 30, 50, and 100 accessions, respectively; GEBV-GD-30, -50, -100: subsets of the top two accessions plus eight accessions chosen from the remainder of the candidate sets composed of the top 30, 50, and 100 accessions, respectively, with the maximal D-scores.

[2] $\overline{GEBV}_P$: average GEBV of the 10 selected parental lines.

[3] $\overline{GEBV}_{F_{10}}$: average GEBV of 2,700 $F_{10}$ RILs.

[4] Lowercase letters indicate significant differences among the strategies for a given trait ($P < 0.01$; LSD test).

including GEBV-O, GEBV-GD-30, -50, and -100 performed equally well for PH but performed significantly better than GD-O-30, -50, and -100.

It is well known that Dataset I contains more genomic diversity than Dataset II since it consists of five subpopulations and one admixed group. The higher genomic diversity of Dataset I could result in a bigger difference between GEBV-GD-30/50/100 strategies and the GEBV-O strategy for some traits. For example, the difference in the GEBV averages among the best $F_{10}$ RILs between GEBV-GD-50 and GEBV-O was approximately -9.06 days for FTAA and -2.55 days for FTAF in Dataset I (Table 1), but the corresponding difference was only -0.09 days for FT in Dataset II (Table 2). However, the flowering time is very sensitive to environmental

conditions, implying that genomic diversity cannot solely amount to the differences in results between these two datasets. More interestingly, the higher genomic diversity of Dataset I could lead to a larger genetic gain for a specific trait. The average genetic gain using the seven strategies for PH in Dataset I was -42.15 cm (Table 4); however, the corresponding mean in Dataset II was only -13.79 cm (Table 5).

The average GEBV of the best $F_{10}$ RILs for YLD was the highest using the GD-O-100 strategy on Dataset II (Table 2). However, the corresponding GEBV averages for two yield components, FPP and PNPP, were the lowest in Dataset I (Table 1). This is possible because the analysis results were based on two different collections of rice lines. There is little diversity among the RILs in Dataset II; therefore, the difference in the average GEBV for YLD among the strategies seems to be negligible. Note that the LSD test revealed only two significance groups in YLD. Nonetheless, the results of FPP and PNPP analysis using the GD-O-100 strategy in Dataset I appear to be reasonable.

Apparently, the number of accessions fixed in the proposed strategies seemed to be arbitrary, similar to the selection of 10 parental lines, retaining the top 2 accessions, and searching 10 or another 8 accessions from the three candidate sets composed of the top 30, 50, and 100 accessions, respectively. A user certainly can adjust these numbers in the strategies according to their own study. Additionally, historical phenotypic data were required to build the GP model. If the historical phenotypic data are not available, then a pilot experiment is needed to phenotype a set of accessions, which can be determined using an optimization algorithm [22]. Two R functions used to perform the proposed procedure for selecting parental lines are provided in S1 File.

We addressed the issue that incorporating genomic diversity into the conventional truncation selection could improve the likelihood of identifying superior parental lines. More importantly, we showed that combining GP with Monte Carlo simulation could help breeders to discover superior parental lines before conducting field experiments. It is well known that phenotype is affected by the genotype (G), environment (E), and G × E interaction. In reality, environment can have a significant impact on the performance of progeny populations during the growth period of each generation until reaching the $F_{10}$ generation. Thus, parental lines selected from our simulation study may not perform as expected. Therefore, conducting field experiments to validate our study would be worthwhile. As mentioned earlier, the works of Gaynor et al. [13] and Yao et al. [14] support that the strategy for selecting better parental lines through GP with Monte Carlo simulation should prove useful in plant breeding. The study of Vanavermaete et al. [15] supports our theory that considering both GEBV and genomic diversity in parental selection is a promising strategy.

In this study, we focused on single-trait selection; therefore, the proposed approach could select different parental lines for different target traits. In practice, it is desirable to extend the approach to multiple-trait selection. A straightforward extension is to apply a selection index incorporating multiple traits, and then treat the selection index as a new target trait for the current single-trait approach. Another possible modification is to directly implement multiple-trait GP models, and then use an appropriate selection index for evaluating candidate accessions. Jia and Jannink [24], Hayashi and Iwata [25], and Guo et al. [26] have shown that multiple-trait GP models provide better prediction accuracy than single-trait GP models for a low-heritability trait, which shows strong correlation with a high-heritability trait. We will present results of the multiple-trait approach in a future communication.

## Conclusion

Combining GP with Monte Carlo simulation can serve as a practical means of detecting superior parents for crop pre-breeding programs. Different strategies can be implemented to

identify a set of superior parental lines from a candidate population. The strategy that considers only GEBV will have a higher starting average GEBV among selected parental lines, but it may lead to a less genetic gain. On the other hand, strategies that consider genomic diversity only can retain greater genomic diversity but cannot simultaneously have a favorable starting GEBV average, and therefore may not produce RILs with satisfactory performance. Strategies that consider both GEBV and genomic diversity balance the starting GEBV average and genomic diversity among parental lines; these strategies show satisfactory genetic gain and produce high-performing RILs.

## Supporting information

**S1 File. R functions.**
(DOCX)

## Acknowledgments

We thank an anonymous reviewer for providing constructive comments, which helped us improve both the content and presentation of the manuscript.

## Author Contributions

**Conceptualization:** Chen-Tuo Liao.

**Data curation:** Ping-Yuan Chung.

**Investigation:** Ping-Yuan Chung.

**Project administration:** Chen-Tuo Liao.

**Software:** Ping-Yuan Chung.

**Supervision:** Chen-Tuo Liao.

**Validation:** Ping-Yuan Chung.

**Writing – original draft:** Ping-Yuan Chung, Chen-Tuo Liao.

**Writing – review & editing:** Chen-Tuo Liao.

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
