## [Decision Letter · Decision Letter 0]

13 Oct 2020

PONE-D-20-25857

Identify Superior Lines for Biparental Crossing via Genomic Prediction

PLOS ONE

Dear Dr. Liao,

Thank you for submitting your manuscript to PLOS ONE. After careful consideration, we feel that it has merit but does not fully meet PLOS ONE’s publication criteria as it currently stands. Therefore, we invite you to submit a revised version of the manuscript that addresses the points raised during the review process.

I had very difficulty time to obtain review comments from the second reviewer in a timely fashion. I believe that the first reviewer's comments are reasonable, thus, I recommend you to make revisions per the comments.

We look forward to receiving your revised manuscript.

Kind regards,

David D Fang, Ph.D.

Academic Editor

PLOS ONE

Journal Requirements:

'This research was supported by the Ministry of Science and Technology, Taiwan (grant number: MOST 108-2118-M-002-002-MY2).'

We note that you have provided funding information that is not currently declared in your Funding Statement, i.e. grant number. However, funding information should not appear in the Acknowledgments section or other areas of your manuscript. We will only publish funding information present in the Funding Statement section of the online submission form.

'MOST (Taiwan) funded the master study of PY, but the funder had no role in study design, data collection and analysis, decision to publish, or preparation of the manuscript. '

Reviewers' comments:

Reviewer's Responses to Questions

**Comments to the Author**

1. Is the manuscript technically sound, and do the data support the conclusions?

Reviewer #1: Partly

2. Has the statistical analysis been performed appropriately and rigorously? 

Reviewer #1: No

3. Have the authors made all data underlying the findings in their manuscript fully available?

Reviewer #1: No

4. Is the manuscript presented in an intelligible fashion and written in standard English?

Reviewer #1: No

5. Review Comments to the Author

Reviewer #1: The strategy for selecting better parental line through genomic prediction (GP) is an excellent idea. This will obviously help the breeder to select the best parents efficiently for getting higher genetic gain. The authors tried to explore GP to select best parental lines for bi-parental crossing based on genomic estimated breeding value (GEBV) couple with genomic diversity (GD) using third party data and simulation study using computer. However, I do have several issue regarding this study and manuscript (MS). I would recommend PlosOne to accept this MS with major revision. Please see below my concern and comments.

1. Authors compared the performance of seven strategies for selecting better parental lines by using the actual average value of traits of top 10 lines. They didn’t do any statistical analysis for comparing the significance of the differences. They need to do some kind of statically analysis for any kind of comparing results in this MS.

2. This is a simulation study. In reality, they may not perform similarly since environment will play a big role on the performance of the progeny during the growing cycle of each of the generation up to F10. It would be worthy, if authors can include some kinds of validation work from the real situation.

3. Authors wrote in the line# 320-23, “Apparently, the numbers of accessions fixed in the proposed strategies seem to be a little arbitrary, such as those of selecting 10 parental lines, retaining the top 2 accessions, and searching 10 or another 8 accessions from the three candidate sets composed of the top 30, 50 and 100 accessions, respectively.” My question is why they select it arbitrary instead selecting top based on actual trait value or doing some statistical analysis. Another concern, how they select the top lines for different traits as they could be different for different traits.

4. They need to rewrite the whole discussion part since many of the statements are redundant with elsewhere especially results section in this MS. They need to include more supporting literature of their outcome. Also they did not explain well of their interesting results.

5. Authors wrote, “An R function for performing the proposed procedure of selecting parental lines is available from the authors upon request”. I would request to authors to include those as supplementary materials for the convenient to readers.

6. Abstract need to rewrite since more than half contains is background. Only four lines (~15%) results and implication. They need to write more results and outcome from this study.

7. Some of the sentences are totally identical in the more than one sections.

8. They need to define first the abbreviation before use in the main body other than abstract.

9. Authors referred one publication in the MS (line# 58) for those who will interest to learn more. I believe that it is unnecessary.

10. In the dataset 1, authors used only one third of the available SNPs. My question is how did they select the SNPs and what was the distribution of SNPs in the genome. Please put some numbers.

11. Authors wrote that “we then imputed a missing SNP marker from its corresponding major homozygous alleles.” I have no idea how could they imputed a missing marker using alleles data.

12. Why and how did they reduce the number of progeny (328 from 363) and SNPs (10,772 out of 73, 147) in the dataset 2.

13. What is the “e” stand for in the equation that measured the recombination rate?

14. The GEBV average of F10 RILs for yield was the highest in the GD-0-100. However, GEBV for all other yield components were lowest. How is it possible? What are the possible explanation? Please include in the discussion.

15. Authors wrote that “the GEBV averages of the best selected parental lines by the strategies can be ranked as GEBV-O = GEBV-GD-30 = GEBV-GD-50 = GEBV-GD-100 > GD-O-30 > GD-O-50 > GD-O-100 in decreasing desirability”. Results of the corresponding figures 2 and 3 do not support this statement for the all traits.

16. English language and grammar need to improve. Some references do not have required information such as page number.

17. In table 2, ranking for PH is not correct.

6. PLOS authors have the option to publish the peer review history of their article (what does this mean?). If published, this will include your full peer review and any attached files.

Reviewer #1: No

---

## [Author Response · Author response to Decision Letter 0]

13 Nov 2020

We have revised the manuscript according to the decision letter.

---

## [Editor Report · Decision Letter 1]

17 Nov 2020

Identification of superior parental lines for biparental crossing via genomic prediction

PONE-D-20-25857R1

Dear Dr. Liao,

We’re pleased to inform you that your manuscript has been judged scientifically suitable for publication and will be formally accepted for publication once it meets all outstanding technical requirements.

Kind regards,

David D Fang, Ph.D.

Academic Editor

PLOS ONE
---

## [Editor Report · Acceptance letter]

23 Nov 2020

PONE-D-20-25857R1 

Identification of superior parental lines for biparental crossing via genomic prediction 

Dear Dr. Liao:

I'm pleased to inform you that your manuscript has been deemed suitable for publication in PLOS ONE. Congratulations! Your manuscript is now with our production department. 

Kind regards, 

on behalf of

Dr. David D Fang 

Academic Editor

PLOS ONE